# Knowledge and Attitude on Childhood Vaccination among Healthcare Workers in Hospital Universiti Sains Malaysia

**DOI:** 10.3390/vaccines10071017

**Published:** 2022-06-24

**Authors:** Ahmad Faiq Mukhtar, Azidah Abdul Kadir, Norhayati Mohd Noor, Ahmad Hazim Mohammad

**Affiliations:** 1Department of Family Medicine, School of Medical Sciences, Universiti Sains Malaysia, Kubang Kerian 16150, Kelantan, Malaysia; afaiqmukhtar@student.usm.my (A.F.M.); hayatikk@usm.my (N.M.N.); 2Hospital Universiti Sains Malaysia, Jalan Raja Perempuan Zainab II, Kubang Kerian, Kota Bharu 16150, Kelantan, Malaysia; 3Department of Community and Family Medicine, Universiti Malaysia Sabah, Kota Kinabalu 88400, Sabah, Malaysia; azimad88@gmail.com

**Keywords:** knowledge, attitude, healthcare workers, childhood vaccination

## Abstract

(1) Background: Vaccine hesitancy is recognized as an important issue globally and healthcare workers (HCWs) have a powerful influence on the public. Recent studies have reported that there are increasing numbers of vaccine hesitancies among HCWs. This study was conducted to assess the knowledge and attitudes on childhood vaccinations among HCWs in Hospital Universiti Sains Malaysia (HUSM). (2) Methods: This is a cross-sectional study conducted among one hundred and ninety-eight HCWs in HUSM, Kubang Kerian, Kelantan who were selected via convenient sampling. Data on their socio-demographic details, working experience, and main source of information regarding childhood vaccination were collected. A validated, Malay version of the knowledge and attitude on childhood vaccination (KACV) questionnaire was used during the study. (3) Results: Female (OR (95% CI):3.15, (1.39, 7.12), *p* < 0.05) and a higher education level (degree and above) (OR (95% CI): 2.36 (1.14, 4.89), *p* < 0.05) are significantly associated with good knowledge. Respondents with a history of side effects of the vaccines among their relatives were about 66% less likely to have good knowledge (OR (95% CI): 0.342 (0.16, 0.73), *p* < 0.05). A positive attitude towards childhood vaccination was significantly associated with a higher level of education participants, who had significantly better knowledge than participants with a lower education level (OR (95% CI): 3.81, (1.92, 7.57), *p* < 0.001). On the contrary, participants having direct contact with patients were less likely to have a good attitude towards childhood vaccination (OR (95% CI): 0.207 (0.043, 0.10), *p* < 0.05), and those with a history of severe side effects of the vaccines among their relatives were also significantly associated with a poor attitude towards childhood vaccination (OR (95% CI: 0.342 (0.16, 0.76), *p* < 0.05).; (4) Conclusions: The survey findings showed a good level of knowledge and a good attitude of participants towards childhood vaccination. Good knowledge is important for the HCWs to have a favourable attitude to educate the general population on childhood vaccination.

## 1. Introduction

Immunization has changed medicine by lowering mortality and morbidity and drastically altering the epidemiology of infectious illnesses. Despite the positive results of the immunization program, a rising percentage of people appear to believe it is hazardous and unneeded. ‘Vaccine hesitancy’ is a word used to characterize individuals who have varied degrees of hesitation regarding specific vaccines or immunization in general, which has recently received increased attention. These people may avoid specific vaccines, postpone vaccinations, or take vaccines but are unsure about it [1]. Vaccine hesitancy has been identified by the World Health Organization as one of the major global health issues for 2019. However, information on this topic is scarce in middle- and low-income nations. This is a significant issue all around the world, particularly in the United States of America. It has resulted in reduced vaccination rates among children and may produce epidemics of diseases such as measles, pertussis, and mumps [2,3]. According to a 2016 study that looked at the link between vaccine delay, refusal, and the epidemiology of measles and pertussis epidemics in the United States, 574 of the 970 measles cases were unvaccinated [3]. They also discovered that the rise in vaccine hesitancy was linked to a higher risk of measles infection in both those who refuse immunizations and those who had been fully vaccinated [3]. In 2012, the World Health Organization (WHO) approved the Global Vaccine Action Plan (GVAP) 2011–2020, which aims to reach 90% immunization coverage for diphtheria, tetanus, and pertussis (DPT) in all countries. Despite the fact that most nations’ childhood vaccination coverage has improved since 2000, many Southeast Asian countries have begun to observe declining trends [4]. In 2018, Indonesia reported that vaccination coverage for children under the age of 12 years was 57.9%, with a downward trend since 2014 [5]. In Malaysia, 86.4% of children were verified as having received complete primary vaccination by the age of 12 months, with an additional 8.9% self-reporting that their child received complete primary vaccination [6], which is less than the 95% target for herd immunity [7]. According to the findings of the National Health and Morbidity Survey 2016, the general prevalence of children aged 12 to 23 months who completed their main immunization was over 90%. However, statistics show that the number of Malaysian parents who choose not to vaccinate their children has increased, particularly among parents of children under 2 years of age, from 637 in 2013 to 1603 in 2016 [6]. Despite a vast range of safe and efficient vaccinations in use worldwide, this rising tendency indicates that the public wants greater assurances about vaccination or immunization [8].

Vaccine acceptance is defined as “the degree to which individuals accept, question, or refuse vaccination.” It is a crucial factor for vaccine uptake rate, and consequently vaccine distribution success [9]. According to WHO in 2019, debates over the effectiveness and safety of vaccinations, in general, have increased in number around the world, posing a serious challenge to global public health [10]. At the same time, several studies have found that accepting, postponing, or refusing vaccination is associated with a number of environmental, social, individual, and vaccine-specific factors [11,12]. Vaccination confidence, which refers to attitudes and beliefs about the advantages and safety of vaccines, as well as trust in vaccine providers, such as healthcare workers (HCWs), health authorities, and policymakers, is an important psychological correlate of vaccination behaviour. Previous studies further suggest that HCWs with higher confidence in vaccinations and vaccine providers are more willing to take the vaccines themselves and to recommend vaccines to patients [13,14]. HCWs’ vaccination confidence to perceive the benefit and safety of vaccines and trust in health professionals, their decisions to accept vaccines for themselves and their children, and their willingness are important for HCWs to recommend vaccines to their patients. HCWs’ advise and recommendation behaviour seems to play an important role in vaccination decisions.

Healthcare workers are considered as the most trusted source of vaccine-related information for patients. They are in the best position to understand issues of hesitancy among patients, respond to their worries and concerns, and find ways to explain to them the benefits of vaccination. However, more and more studies show that HCWs themselves, including those who provide vaccination to patients, provide wrong information or even advise patients to prevent taking vaccines [15]. Almost all these studies in Europe focused on HCWs’ attitudes and concerns about seasonal and pandemic influenza vaccines. Many of the studies found that HCWs were not vaccinated against influenza because they had not had time [16,17], were not at risk of influenza [18], felt healthy or had not had the vaccine prescribed [19,20], or had concerns about vaccine safety and efficacy [21]. A recently published study showed that 16% to 43% of French family doctors admitted not having or only sometimes recommending a specific vaccine for their patients [15]. Concurrently, there has been a lack of information among professional healthcare providers, leading to substantial scepticism regarding the efficacy and safety of vaccinations, potentially leading to widespread under-utilization. This attitude of mistrust manifested itself clearly after the recent influenza pandemic when HCWs showed little support for vaccination efforts, even though flu vaccination is recommended for all at-risk populations, including HCWs [22,23]. Not all HCWs favour vaccinations, as seen by the numerous websites and healthcare worker associations that openly opposed vaccines [24].

A study was performed in Croatia, France, Greece, and Romania to investigate possible vaccination concerns among HCWs. The findings revealed that vaccination hesitancy exists among vaccine providers in all four nations. The fear of vaccine adverse effects was the most common concern across all countries [25]. In another study conducted in Spain, despite the fact that the majority of paediatric nurses intended to vaccinate their own children, about one-third expressed vaccine hesitancy, primarily due to concerns about HPV and varicella vaccines, as well as other misconceptions [26]. In Asia, the discussion on vaccine hesitancy is limited; however, certain studies are progressing. Vaccine hesitancy, especially among HCWs, jeopardizes decades of progress in decreasing the burden of infectious diseases that have troubled humankind for generations. Myths and disinformation about vaccination can only be dispelled by collaboration between paediatricians, family doctors, parents, public health officials, governments, and civil society.

This survey was designed to determine the knowledge, attitude, and associated factors for childhood vaccinations of HCWs at the Hospital Universiti Sains Malaysia (HUSM). The goal of the single tertiary centre study was to look at childhood vaccinations of HCWs in the hospital, their worries about the vaccines, and any other variables that might be causing hesitancy. We also wanted to find out if there were any independent determinants of vaccination willingness. The findings may affect the acceptance of vaccination at the primary care level as HUSM is a referral hospital and it may prevent the spreading of wrong information on childhood vaccination from HCWs to society.

## 2. Materials and Methods

### 2.1. Study Setting and Study Population

A cross-sectional, descriptive study was conducted in March 2021 in HUSM. The study was conducted in outpatient clinics, laboratories, and wards. Inclusive criteria for the study were all types of professions of HCWs (doctors, dentists, nurses, health assistants, laboratory technicians, radiology technicians, dietitians, and physiotherapists) including postgraduate students, permanent and contract workers, those who have USM staff emails, Malaysian citizens, and working experience in HUSM for at least six months. Participants were excluded if they were on long leave (maternal leave/sabbatical leave) during the survey.

Convenient sampling was used. The sample size was calculated for the variable types of the profession by comparing two means. Using the standard deviation of 1.07 among professionals from a previous study [27], a detectable difference of 0.5, power of 80%, and the minimum sample size is 146 HCWs. After considering a non-response rate of 30%, the calculated sample size was 190 HCWs. The questionnaire was distributed by the research assistant. The data collection in each location was distributed randomly, depending on the availability of HCWs. The participants were invited to participate in the survey after a brief explanation of the study in the initial page questionnaire. The questionnaire was self-administered, and it took about 10 min to be completed.

### 2.2. Questionnaire KACV

A validated Malay version of the questionnaire on knowledge and attitude on childhood vaccination was used in this study [28]. This questionnaire was developed specifically for HCWs and a psychometric validation study was conducted in the School of Health Sciences, Universiti Sains Malaysia among 110 participants. The preliminary KACV showed a high item Content Validity Index and Face Validity Index. The final questionnaire consists of 10 items for knowledge and 15 items for attitude. The Cronbach alpha for the knowledge and attitude section were 0.896 and 0.861, respectively [29]. The knowledge questions consisted of Yes/No/Unsure response options. The correct answer was provided as “1” and a wrong or unsure answer was provided a score of “0”. The total knowledge scores ranged from 0 to 30. The participant’s responses to the attitudes statement were measured by using a 5-point Likert scale.

Based on the modified Bloom’s cut-off score, the level of knowledge was classified into low level (less than 90%; 0–26 scores) and high knowledge (90% and above; more than 27 scores) [30]. The modified Bloom’s cut-off score of 90% was used because the HCWs are considered the main source of knowledge for the public and was more exposed to medical knowledge. The overall level of attitude was categorized using the original Bloom’s cut-off point. A mean score of 60 (80% or more) was considered a positive attitude while a score of less than 60 was considered a negative attitude [31].

### 2.3. Statistical Analysis

Data analysis was performed by using Statistical Package for Social Sciences (SPSS) (V 27). Frequencies were generated for the sociodemographic characteristics of the participants. Descriptive analysis was used to summarize the results as frequencies and percentages frequencies of the PACV items were calculated. A normality test was based on the Shapiro–Wilk test for age, duration of working, score for knowledge and attitude. The Spearman correlation was run to assess the correlations between knowledge and attitude scores. Univariable analyses were performed to identify associated factors of knowledge and attitude. Multivariable analyses were run by using Hosmer–Lameshow test to determine associated factors for good knowledge and attitude. A *p* value < 0.05 was considered statistically significant.

### 2.4. Ethical Consideration

This study was approved by the Human Research Ethics Committee of Universiti Sains Malaysia (JEPeM) (Ref USM/JEPeM/20050250). The informed consent was obtained first from HCWs before proceeding with the questionnaire. The information is anonymous and confidential to the public. The response obtained is anonymous. Only research team members can access the data. Confidentiality of the data is strictly maintained. Data are presented as grouped data and do not identify the participant individually or the involved participants.

## 3. Results

### 3.1. Descriptive Statistics: Demographic Information, Main Source of Information and Personal Experience of Participants, Participants’ Knowledge and Attitude Score

Of 213 individuals invited during the study, 198 respondents completed the questionnaire. This provided a response rate of 92.9%. Among the respondents, 82.8% were females and 17.2% were males. The mean age was 39.4 years. Most of the respondents were Malay (92.4%) and married (88.4%). The information about demographic details, the main source of information, and the personal experience of the respondents are presented in Table 1.

A total of 68.7% of participants have good knowledge about childhood vaccination. The results showed that the participants know the functions of measles, mumps, and rubella vaccines which are for reducing the risk of measles, mumps, and rubella infection (93.9%). It also showed that participants were aware children who were breastfed exclusively still required vaccination (90.4%) and a vaccination was necessary for the natural immune system of children to develop a defence against vaccine-preventable diseases (90.4%). Respondents’ knowledge about herd immunity was very encouraging (88.4%) whereby children need to be vaccinated even though the other children were already vaccinated, and most respondents agreed that the polio vaccine was needed even though polio disease no longer exists (78.3%). The results for knowledge of childhood vaccination are summarized in Figure 1.

A total of 117 (59.1%) participants showed a good attitude towards childhood vaccination. The results showed that 69.7% of respondents agree to make it mandatory to vaccinate children and 68.7% of participants are confident the vaccines were safe. It also showed that participants were aware many dangerous infectious diseases could be prevented through vaccination (66.7%) and they believed the health information about childhood vaccination from health personnels (64.6%). More than half of respondents were confident with the suitability of the childhood vaccination schedule set by the Ministry of Health (59.1%) and the information that was obtained from the internet about the vaccination should be investigated first (57.1%). On the other hand, 67.2% of participants strongly disagree that giving vaccines to children was unnecessary (Table 2).

Females are significantly associated with a good knowledge, as female respondents were more knowledgeable about childhood vaccination rather than male respondents (OR (95% CI):3.15, (1.39, 7.12), *p* < 0.05). The results also showed that higher education level (degree and above) participants had significantly higher knowledge than lower education level participants (diploma and below) (OR (95% CI): 2.36 (1.14, 4.89), *p* < 0.05).

Respondents with a history of side effects from vaccines among their relatives were about 66% less likely to have good knowledge (OR (95% CI): 0.342 (0.16, 0.73), *p* < 0.05) than those without a history of side effects of the vaccines (Table 3). The sensitivity was 87.5%. The specificity is 43.5% providing an overall percentage of 73.8% with a Hosmer–Lameshow goodness fit for statistics (*p* = 0.831).

### 3.2. Univariate and Multivariate Analyses and the Associated Factors for Good Knowledge and Positive Attitude

A positive attitude towards childhood vaccination was significantly associated with a higher level of educated participants with a significantly better knowledge than participants with a lower education level (OR (95% CI): 3.81, (1.92, 7.57), *p* < 0.001). On the contrary, participants having direct contact with patients were less likely to have a good attitude (OR (95% CI): 0.207 (0.043, 0.10), *p* < 0.05), and those with a history of severe side effects of the vaccines among their relatives were also significantly associated with a poor attitude (OR (95% CI: 0.342 (0.16, 0.76), *p* < 0.05) (Table 3). The sensitivity for attitude is 93.2% and the specificity is 24.7%, providing an overall percentage of 65.2% with a Hosmer–Lameshow goodness for statistics (*p* = 0.928).

## 4. Discussion

The data of the present study showed in general 68.7% of the respondents were knowledgeable and aware of childhood vaccination. Healthcare workers with a good level of knowledge were more likely to be observed in those with a higher level of education (degree and above) and female. There is a clear link between those who had a negative vaccination experience and those who have less knowledge about vaccination and a negative attitude toward childhood vaccination. A substantial proportion of HCWs displayed a good attitude towards childhood vaccination. The majority of the attitude-related statements elicited positive responses from the participants. A majority agreed to make vaccination of children mandatory and were confident with the safety of vaccines provided to children. Healthcare workers with a higher level of education (degree and above) were also more likely to be observed to have a good attitude. In contrast, participants having direct contact with patients were less likely to have a good attitude.

### 4.1. Sociodemographic Factors

It was expected that the score of knowledge and attitude would be high. This was contributed to by the sociodemographic of the participants. The participants consist of HUSM staff from various clinics and departments with different types of occupations. The majority of the participants are paramedics (nurses and medical assistants) and most of the respondents were not directly involved in vaccinating children. HUSM is a tertiary centre in which most of the immunization program is run by primary care centres including health clinic and general practitioners and do not have enough exposure on childhood vaccinations. However, 54% of the respondents have experience working in paediatric wards. In general, those with a medical background or working experience as healthcare providers should have a good knowledge and attitude on childhood vaccination. The score may be different if the study was conducted in primary care settings. A similar study should be carried out at the primary care level to obtain a general score of knowledge and attitude on childhood vaccination.

The majority of the respondents are female. These findings reflect the epidemiological data of the HCWs in the hospital where most of them are females. Most of the respondents that participated in the study are nurses, doctors, and laboratory technicians which consist mainly of the female. Since a majority of respondents are females, the findings might not be representative of the general population. However, after analysing data with multiple logistic regression, we already adjusted and controlled other cofounders including the interaction of all factors.

### 4.2. Knowledge and Associated Factors

This study enables us to comprehend the level of knowledge and attitudes on childhood vaccinations among a group of HCWs in HUSM. The data of the present study showed, in general, 68.7% of the respondents were knowledgeable and aware of childhood vaccinations. Healthcare workers play a crucial role in vaccinations, especially for children. They should have sufficient knowledge to correctly educate the general population and the most susceptible and fragile groups.

Several previous studies among different groups of individuals conducted in the same geographic area have underlined that HCWs are the most important and trusted source of information on vaccine-preventable infectious diseases [24,25,26,27]. Thus, the poor level of knowledge is a serious issue because HCWs may place the population at risk of contracting vaccine-preventable diseases resulting in the transmission of diseases. HCWs with a good level of knowledge were more likely to be observed in those with a higher level of education (degree and above) and female. Previous investigations have found that those with a higher level of education show better knowledge of vaccinations [31,32]. Those studies determine the knowledge and acceptance among HCWs in hospitals in China and Saudi Arabia towards hepatitis B and influenza vaccinations. To dispel HCWs’ misconceptions regarding the appropriate knowledge, awareness programmes are required. Following the establishment of a mandatory vaccination policy there was a good uptake of influenza vaccine during the 2014/15 season, especially among doctors. This may be since more exposure concerning vaccinations has been included during their study period. Thus, more effort should be taken into an action to strengthen the knowledge training curriculum, and improve the tools to understand the benefits of vaccines among those with poor knowledge. There is a clear link between those who had a negative vaccination experience and those who have less vaccination knowledge and a negative attitude about vaccination. This is due to the side effects that are associated with a vaccination to be considered as severe enough to make vaccines unacceptably dangerous [33]. It was suggested vaccine hesitancy was higher for those who experience or witnessed the negative side effects after taking a vaccine. To improve the current situation, we must define both the risks and the benefits of individual vaccines so that the public can understand the rationale for vaccine recommendations. The key to regaining public trust in vaccines is a credible, consistent, and unified message developed from the private and public sectors that directly addresses public concerns. Those who had a negative experience with vaccinations should have a more individualized approach so that their unfavourable perception can be modified through appropriate and proper understanding. These issues can be addressed through targeted awareness campaigns and education to combat misconceptions and concerns regarding vaccine safety and efficacy and increase confidence in vaccines.

A substantial number of participants believed that the measles vaccine can cause autism and the side effects of the polio vaccine can cause paralysis of the legs. A study on measles, mumps, and rubella (MMR) vaccines causing autism was published in 1998, but was subsequently retracted by the journal. Several epidemiologic studies have found no link between MMR vaccination and autism, including one that indicated that the vaccine was not linked to an elevated risk of autism even in high-risk children with autism found in their older siblings [34]. On the other hand, the major risks of the oral polio vaccine (OPV) are the associations with vaccine-associated paralytic poliomyelitis cases (VAPP) and the emergence of vaccine derived polioviruses strains. However, most countries, including Malaysia, switched the schedule of vaccination by using inactivated polio vaccine (IPV) instead of OPV because it provides no risk of vaccine-related disease [35]. Quite a number of participants still believe in these side effects posed by the vaccines. Despite strong evidence of its safety, some parents are still hesitant to accept MMR vaccination of their children. Therefore, health-care providers must have the correct understanding to play a vital role in maintaining confidence in vaccinations and preventing suffering, disability, and death from vaccine-preventable diseases [34]. Educational programmes are crucial to improve the HCWs’ knowledge about childhood vaccination, for example, more continuous medical education and courses regarding childhood vaccinations.

### 4.3. Attitude and Associated Factors

A substantial proportion of HCWs displayed a good attitude towards childhood vaccination. The majority of the attitude-related statements elicited positive responses from the participants. A majority agreed to make it mandatory to vaccinate children and were confident with the safety of vaccines administered to children. On the contrary, about half of the participants agreed that children now receive too many types of vaccine injections. This is worrisome since these numbers significantly show a negative attitude that is not supposed to be present among HCWs. A study in Spain identifies low levels of rejection against the new COVID-19 vaccine, as identified among HCWs. However, even being at a higher risk, healthcare professionals did not show higher positive attitudes toward vaccines. Furthermore, a refusal percentage for vaccination was higher among healthcare professionals compared with non-healthcare professionals [36]. This apparent issue should be addressed for further intervention to convey a correct understanding of vaccinations for a better attitude, in our case, on childhood vaccination. Again, we recognized knowledge as a significant factor inducing good attitudes toward childhood vaccination. HCWs with a higher level of education (degree and above) were more likely to be observed to have a good attitude. In contrast, participants having direct contact with patients were less likely to have a good attitude. This was surprising, considering that previous research had found a strong link between experience, knowledge, and attitude [37]. The study identified low pertussis and influenza vaccination among pregnant women and their HCWs due to inadequate knowledge of immunization guidelines among HCWs, lack of insight or poor attitude on the need to obtain the vaccines among HCWs, and poor communication are the probable causes of the low uptake. Being a healthcare worker was associated with having more knowledge, which led to a more positive attitude. This may be because healthcare personnel are more likely to have encountered cases of vaccine-preventable infectious illnesses. A recent study about COVID-19 vaccination among HCWs in Baltimore, Maryland shows the positive attitude on the vaccine resulting in an overall improvement in the HCWs’ mental health, well-being and increased comfort in caring for COVID-19 and non-COVID-19 patients [38]. The study highlights the importance of widespread vaccination of HCWs from personal and professional perspectives and may be transmissible to the general population.

Further studies need to be performed to determine the possible causes of these unexpected results. Finally, more than half of HCWs with a history of severe side effects of the vaccines among their relatives were less likely to have a good attitude. This is owing to a widespread perception that any potential dangers associated with a vaccination are severe enough to consider vaccinations unacceptably unsafe [30]. Despite scientific facts, negative concerns regarding vaccines can harm children and jeopardize vaccination efforts. As a result, continuous education programmes for HCWs should be made available together with reliable data on the effects of vaccinations, particularly accurate data on adverse effects.

The strength of this study was validated by the Malay version questionnaire to assess the sociodemographic, knowledge, and attitude of HCWs in HUSM [28]. A panel of experienced statistician, paediatricians, and family medicine specialists revised the questionnaire. This ensured that the data collected was valid and reliable for the study. Other than that, the questionnaire was based on the culture and sociodemographic of general population in Malaysia. Thus, it can be used to assess knowledge and attitude on childhood vaccinations among HCWs in Malaysia. The study can be expended to the whole country.

### 4.4. Limitations

This study also has several limitations that should be considered to make an appropriate interpretation of the findings. The first limitation is that the questionnaire was self-administered, and there was no attempt to independently verify the respondents’ information; hence, social desirability, recall, and answer biases are possible. To reduce the risk of bias in responding, the survey was conducted anonymously with no identifying data collected. Second, the sample was formed of participants from a single geographic location mainly consisting of the Malay population; thus, the results’ generalizability to the rest of the country needs to be proven. Another limitation is that this study was undertaken while the COVID-19 epidemic in Malaysia was still active and worsening. There was some debate about the COVID-19 vaccination in terms of efficacy and safety. This may have a detrimental impact on the participants’ views about vaccinations to some extent. One more limitation is the majority of respondents were females. This is due to some limitations in data collection during the peak of COVID-19 pandemic with the majority of the staff in the hospital being females. This might not be representative of the general population. The study may be conducted in the future with more favourable proportion between male and female respondents, as some limitations during COVID-19 pandemic have been eased.

## 5. Conclusions

The survey findings showed a good level of knowledge and a good attitude of the participants towards childhood vaccination. A good level knowledge is important for the HCWs to have a favourable attitude to educate the general population on childhood vaccination. A negative experience on vaccination was the major factor for poor knowledge which resulted in a negative attitude towards childhood vaccination. These findings highlight the need for increased efforts by policymakers to educate HCWs to enhance their knowledge and attitude on childhood vaccination. To improve the knowledge and to a have a positive attitude toward childhood vaccination among HCWs, we need to develop and implement vaccination communication strategies that address the determinants of vaccine hesitancy. These strategies should provide confidence in childhood vaccination by debunking vaccine myths and rumours, correcting misinformation, and scientifically addressing vaccine safety concerns.

## Figures and Tables

**Figure 1 vaccines-10-01017-f001:**
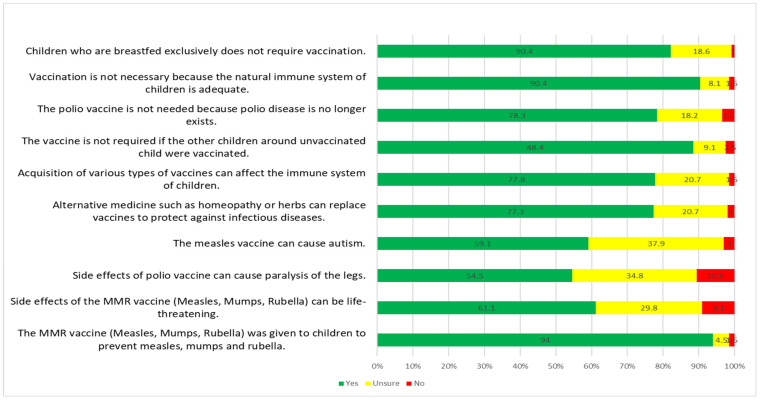
The results for knowledge of childhood vaccination.

**Table 1 vaccines-10-01017-t001:** Demographic information, main source of information, and personal experience of participants (n = 198).

Variables		Total (%)	Median (IQR)
Age (years)	<40	122 (61.6)	37.00 (12.00)
	≥40	76 (38.4)	
Sex	Male	34 (17.2)	
	Female	164 (82.8)	
Race	Malay	183 (92.4)	
	Others	15 (7.6)	
Religion	Islam	184 (92.9)	
	Others	14 (7.1)	
Education	Diploma/Certificate	127 (64.1)	
	Degree/Master	71 (35.9)	
Occupation	Paramedics	92 (46.5)	
	Doctors	34 (17.2)	
	Med Support	72 (36.4)	
Marital	Single	23 (11.6)	
	Married	175 (88.4)	
Child	0	31 (15.7)	
	1–3	100 (50.5)	
	≥4	67 (33.8)	
Service Duration	1–10	76 (38.4)	13.00 (11.25)
(years)	11–20	76 (38.4)	
	≥21	46 (23.2)	
Source	Professional	120 (60.6)	
	Non-Professional	78 (39.4)	
Having direct contact/engagement with patient		184 (92.9)	
Working in paediatric ward/unit		107 (54.0)	
Administering vaccine injection to children		20 (10.1)	
Attended course on child vaccination		26 (13.1)	
Having relatives/friends reported severe side effect of vaccine		37 (18.7)	
Taking additional vaccine for yourself (i.e., influenza and pneumococcal)		96 (48.5)	

**Table 2 vaccines-10-01017-t002:** Participants’ attitude on childhood vaccination.

Attitude Statement	Participants’ Responses n (%)				
	Strongly Agree	Agree	Neutral	Disagree	Strongly Disagree
I am confident the vaccine is given to children are safe.	136 (68.7)	54 (27.3)	6 (3.0)	1 (0.5)	1 (0.5)
I feel children now receive too many types of vaccine injections.	15 (23.7)	43 (21.7)	42 (21.2)	51 (25.8)	47 (23.7)
I believe many dangerous diseases can be prevented through vaccination.	132 (66.7)	54 (27.3)	11 (5.6)	0 (0)	1 (0.5)
I believe it is better to get immunity through infection than vaccination.	10 (5.1)	22 (11.1)	31 (15.7)	62 (31.3)	73 (36.9)
I am worried about a severe allergic reaction due to the vaccination of children.	11 (5.6)	46 (23.2)	54 (27.3)	57 (28.8)	30 (15.2)
I think childhood vaccinations do more harm than good.	5 (2.5)	7 (3.5)	23 (11.6)	54 (27.3)	109 (55.1)
I agree to make it mandatory to vaccinate children.	138 (69.7)	38 (19.2)	17 (8.6)	2 (1.0)	3 (1.5)
Parents/guardians have the right to refuse vaccination of children.	7 (3.5)	40 (20.2)	53 (26.8)	41 (20.7)	57 (28.8)
I believe the health information about childhood vaccination of health personnel.	128 (64.6)	58 (29.3)	8 (4.0)	2 (1.0)	2 (1.0)
I am confident with the vaccination information obtained through the internet.	25 (12.6)	58 (29.8)	76 (38.4)	26 (13.1)	13 (6.6)
I think the information that is navigable from the internet about the vaccination should be investigated first.	113 (57.1)	52 (26.3)	26 (13.1)	4 (2.0)	3 (1.5)
I believe vaccines are a hidden agenda to benefit drug companies.	3 (1.5)	8 (4.0)	47 (23.7)	58 (29.3)	82 (41.4)
I believe vaccination is not effective in providing protection against infectious diseases.	6 (3.0)	13 (6.6)	22 (11.1)	63 (31.8)	94 (47.5)
I am confident with the suitability of childhood vaccination schedule set by the Ministry of Health.	117 (59.1)	69 (34.8)	9 (4.5)	2 (1.0)	1 (0.5)
I think giving vaccines to children is unnecessary.	3 (1.5)	7 (3.5)	12 (6.1)	43 (21.7)	133 (67.2)

**Table 3 vaccines-10-01017-t003:** Association of variables with knowledge and attitude on childhood vaccination.

		Simple Logistic Regression				Multiple Logistic Regression			
		Knowledge		Attitude		Knowledge		Attitude	
Variables	Distribution	Crude OR (95% CI)	*p* Value	Crude OR (95% CI)	*p* Value	Crude OR (95% CI)	*p* Value	Crude OR (95% CI)	*p* Value
Age (years)	<40	1.00		1.00		-	-	-	-
	≥40	0.612 (0.323, 1.160)	0.132	0.709 (0.397, 1.268)	0.246	-	-	-	-
Sex	Male	1.00		1.00		-	-	-	-
	Female	2.644 (1.243, 5.625)	0.012 *	0.751 (0.348, 1.620)	0.465	3.148 (1.393, 7.118)	0.006 *	-	-
Race	Malay	1.00		1.00		-	-	-	-
	Others	3.171 (0.693, 14.502)	0.137	1.998 (0.613, 6.510)	0.251	-	-	-	-
Religion	Islam	1.00		1.00		-	-	-	-
	Others	2.903 (0.630, 13.386)	0.172	1.799 (0.544, 5.949)	0.336	-	-	-	-
Education	Diploma/Cert	1.00		1.00		-	-	-	-
	Degree/Master	2.193 (1.118, 4.304)	0.022 *	4.039 (2.072, 7.876)	<0.001 *	2.360 (1.139, 4.892)	0.021 *	3.813 (1.922, 7.568)	<0.001 *
Occupation	Paramedics	1.00		1.00		-	-	-	-
	Doctors	2.948 (1.039, 8.363)	0.042 *	4.029 (1.595, 10.173)	0.003 *	-	-	-	-
	Med Support	0.899 (0.471, 1.717)	0.747	1.741 (0.929, 3.263)	0.084	-	-	-	-
Marital	Single	1.00		1.00		-	-	-	-
	Married	1.480 (0.603, 3.630)	0.392	1.375 (0.575, 3.289)	0.474	-	-	-	-
Child	0	1.00		1.00		-	-	-	-
	1–3	0.831 (0.344, 2.003)	0.679	1.075 (0.469, 2.464)	0.864	-	-	-	-
	≥4	0.961 (0.377, 2.450)	0.934	0.691 (0.290, 1.644)	0.403	-	-	-	-
Serve Duration	1–10	1.00		1.00		-	-	-	-
	11–20	0.748 (0.383, 1.461)	0.395	1.320 (0.686, 2.539)	0.406	-	-	-	-
	≥21	1.897 (0.791, 4.549)	0.151	0.793 (0.380, 1.657)	0.538	-	-	-	-
Source	Non-Professional	1.00		1.00		-	-	-	-
	Professional	0.786 (0.422, 1.464)	0.448	0.846 (0.472, 1.514)	0.572	-	-	-	-
Having direct contact/engagement with patient		1.238 (0.397, 3.859)	0.713	0.222 (0.048, 1.018)	0.053	-	-	0.207 (0.043, 0.995)	0.049 *
Working in paediatric ward/unit		1.683 (0.919, 3.082)	0.092	1.263 (0.715, 2.229)	0.422	-	-	-	-
Administering vaccine injection to children		4.576 (1.028, 20.378)	0.046 *	1.699 (0.624, 4.626)	0.300	-	-	-	-
Attended course on child vaccination		0.575 (0.247, 1.337)	0.198	1.360 (0.574, 3.223)	0.485	-	-	-	-
Having relatives/friends reported severe side effect of vaccine		0.300 (0.144, 0.626)	0.001 *	0.297 (0.140, 0.627)	0.001 *	0.342 (0.160, 0.732)	0.006 *	0.342 (0.155, 0.756)	0.008 *
Taking additional vaccine for yourself		0.915 (0.502, 1.670)	0.713	0.941 (0.534, 1.659)	0.833	-	-	-	-

n = 198, * significant, *p* < 0.05.

## Data Availability

The data presented in this study are available on request from the corresponding author. The data are not publicly available due to privacy and confidentiality.

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
