# Peer review of "Knowledge and Attitude on Childhood Vaccination among Healthcare Workers in Hospital Universiti Sains Malaysia"

_vaccines, 2022, doi:10.3390/vaccines10071017_

Round 1

Reviewer 1 Report

Mukhtar et al. conducted a cross-sectional study of healthcare workers' knowledge and attitudes towards childhood vaccination, a novel study and my main concerns are as follows:

The authors found a clear relationship between females and good knowledge, it is clear that the gender ratio of the enrolled men and women was disproportionate, with 82.8% being female, the authors should have corrected for gender and also considered the association between gender and other variables.

Author Response

Thank you for your comments and suggestions. I really appreciate your guidance for this article. I have made all changes as per comments/suggestions. Thank you very much.

Response:

The majority of the respondents are female. These findings reflect the epidemiological data of the HCWs in the hospital where the majority of them are females. Most of the respondents that participated in the study were nurses, doctors, and laboratory technicians which consist mainly of the female gender. Since a majority of respondents were females, the findings might not be representative of the general population. But, after analyzing data with multiple logistic regression we already adjusted and controlled other cofounders including the interaction of all factors.  The study may be conducted in the future with a more favorable proportion of male and female respondents as some limitations during the COVID-19 pandemic were eased.

(Line 259-267)(Line 386-391)

Reviewer 2 Report

the manuscript covers an interesting topic even more important in covid-19 time. Actually, understanding and exploring the knowledge, attitude, and practice of health care workers on vaccine hesitancy/vaccine acceptance is of paramount importance.

in the introduction section, the authors should present both vaccine hesitancy and vaccine acceptance concepts. 

some results can be shown in graph in order to increase the readability and interest of the readers

In the discussion, the first section with general comments/overview of the main results should be added before presenting every single aspect of the results.

the authors should explain the role of the funders, if any, with data collection, analysis, and publication.

Author Response

Thank you for your comments and suggestions. I really appreciate your guidance for this article. I have made all changes as per comments/suggestions. Thank you very much.

Comments:

  1. In the introduction section, the authors should present both vaccine hesitancy and vaccine acceptance concepts.

Response: Vaccine hesitancy and acceptance concepts were added and discussed in the introduction. (Line 70-86)

2. Some results can be shown in graph in order to increase the readability and interest of the readers.

Response: 

The results of knowledge on childhood vaccination are presented in chart.

Figure 1: The results for knowledge of childhood vaccination (page 6)

3. In the discussion, the first section with general comments/overview of the main results should be added before presenting every single aspect of the results.

Response: 

The data of the present study showed, in general, 68.7% of the respondents were knowledgeable and aware about childhood vaccination. HCWs with a good level of knowledge were more likely to be observed in those with a higher level of education (degree and above) and female gender. There is a clear link between those who have had a negative vaccination experience and those who have less knowledge about vaccination and a negative attitude childhood vaccination. A substantial proportion of healthcare workers dis-played a good level of attitudes towards childhood vaccination. The majority of the attitude-related statements elicited positive responses from the participants. Majority agreed to make it mandatory to vaccinate children and were confident with the safety of vaccines given to children. HCWs with a higher level of education (degree and above) were also more likely to be observed to have a good attitude. In contrast, participants having direct contact with patients were less likely to have a good attitude.

(Line 233-244)

4. The authors should explain the role funders, if any, with data collection, analysis, and publication. 

Response: Funding: This research received funding from Geran Penerbitan Sarjana Perubatan (GPSP) by School of Medical Sciences, Universiti Sains Malaysia (USM). The fund helped in data collection and publication of the study. (Line 413-415)

Reviewer 3 Report

An interesting, informative and meaningful manuscript that has clinical merit.  However, there are some writing/editing issues that the authors should consider and address.  Once an abbreviation is assigned to a word or phrase, then use the abbreviation thereafter in the manuscript.  The following are suggestions/comments for the authors.  Line 14, "that there are increasing numbers of vaccine hesitancies among HCWs."  Lines 21 & 22, "...p<0.05] and a higher education level ...".  Line 26, "...education participants, who had significantly better ...".  Line 30, "...associated with a poor attitude ...".  Line 33, "...important for the HCWs to have a ...".  Line 41, "...immunization in general, which has recently ...".  Line 47, "...among children and may produce epidemics of ...".  Lines 50 & 51, "...cases were unvaccinated [3]. They also discovered ...".  Lines 57 & 58, "...reported that vaccination coverage ...".  Line 70, "Healthcare workers (HCWs) are considered ...".  Line 73, "...more studies show that HCWs themselves, including ...".  Line 77, "...Europe focused on HCWs' attitudes and ...".  Line 78, "...studies found that HCWs had not been ...".  Line 82, "...having or only sometimes recommending a specific vaccine for ...".  Lines 86 & 87, "...influenza pandemic when HCWs showed little support ...".  Lines 88 & 89, "...including HCWs [16-17].  Not all HCWs favor vaccinations, as ...".  Line 98, "...hesitancy is limited; however, certain studies how it progressing."  Line 102, "...officials, governments, and a civil society."  Line 104, "...childhood vaccinations of HCWs t the Hospital ...".  Line 110, "...vaccination from HCWs to society."  Lines 114 & 115, "...in March 2021 in HUSM.  the study ...".  Lines 116 & 117, "...of professions of HCWs (doctors, dentists, ...".  Line 124, "among professionals form a previous ...".  Line 125, "...sample size is 146 HCWs."  Line 126, "...sample size was 190 HCWs."  Line 128, "...on the availability of HCWs.  Line 139, "...were 0.896 and 0.861, respectively [22]."  Line 141, "...as "1" and a wrong or unsure answer was given a score of "0"."  Lines 146 & 147, "...because the HCWs are considered a main source of ...". Line 155, "Normality test was based on Shapiro-Wilk ...".  Line 166, "..forst from HCWs before proceeding ...".  Lines 183 & 184, "...Rubella vaccines, which are for reducing the risk of measles, mumps, and rubella infections (93.9%)."  Line 186, "...(90.4%) and a vaccination was necessary for ...".  Line 190, "...vaccine was needed even though polio disease ...".  Line 192, "Females are significantly associated with a good knowledge as ...".  Line 193, "...childhood vaccination rather than male ...".  Lind 194, "...results also showed that a higher education level ...".  Line 197, "...history of side effects from vaccines among their ...".  Line 201, "...goodness fit for statistics (p=0.831)."  Line 237, "...higher level of educated participants with a significantly better...".  Line 241, "...significantly associated with a poor attitude ...".  Line 243, "...the specificity is 24.7%, giving an overall percentage ...".  Line 244, "...goodness for statistics (p=0.928)."   Lines 252 & 253, "...the respondents not directly involved in ...".  Line 255, "...exposure on childhood vaccinations."  Line 257, "...those with a medical background ...".  Line 258, "...should have a good knowledge and attitude ...".  Line 259, "different if the study was conducted in ...".  Line 265, "hood vaccinations among a group of HCWs in HUSM."  Line 267, "childhood vaccinations.  HCWs play a crucial ...".  Line 271, "...have underlined that HCWs are the ...".  Line 273, "...serious issue because HCWs may put ...".  Line 275, "...of diseases.  HCWs with a good level ...".  Line 278, "...knowledge of vaccinations [25, 26]."  Line 279, "...among HCWs in hospitals ...".  Lines 279 & 280, "...towards Hepatitis B and Influenza vaccinations."  Line 285, "...the knowledge training, curriculum and improve ...".  Line 289, "...side effects that are associated with a vaccination ...".  Line 291, "...side effects after taking a vaccine either ...".  Line 304, "...and rubella (MMR) vaccines causing autism ...".  Line 307, "...children with autism found in their older ...".  Line 313, "...these side effects posed by the ...".  Line 318, "improve the HCWs ' knowledge about ...".  Lines 327 & 328, "This is worrisome since these numbers significantly shows a negative attitude that is supposed to be not present among HCWs."  Line 329, "...new COVID-19 vaccine, as identified among ...".  Line 331, "Furthermore, a refusal percentage for vaccination was ...".  Line 334, "...of vaccination for a better attitude, in our ...".  Line 336, "...childhood vaccination.  HCWs with a higher level ...".  Line 346, "...infectious illnesses.  A recent study ...".  Line 348, "vaccine resulting in an overall improvement in the HCWs' mental ...".  Line 352, "Further studies need to be done to ...".  Line 353, "Finally, HCWs with a history of ...".  Line 358, "...education programmes for HCWs should be ...".  Line 361, "...of this study validated the Malay version ...".  Line 364, "This ensured that the data collected was valid and reliable.....".  Line 365, "...the questionnaire was based on the culture ...".  Line 367, "childhood vaccinations among HCWs in Malaysia.  The study can be expanded to the ...".  Line 371, "The first limitation is that questionnaire was ...".  Line 372, "...self-administered, not to attempt independently to verify the respondents'...".  Line 373, "...recall, and answer biases are possible."  Line 380, "...views about vaccinations to some ...".  Line 383, "...knowledge and attitude of the participants ...".  Line 384, "...childhood vaccination.  A good knowledge is important for the HCWs to have a ...".  Line 386, "A negative experience on ...".  Line 388, "...to educate HCWs to enhance ...".  Lines 389 & 390, "...and to have a positive attitude of childhood .,..."

Author Response

Thank you for your comments and suggestions. I really appreciate your guidance for this article. I have made all changes as per comments/suggestions. Thank you very much.

Response: Corrections made as per comments.

Round 2

Reviewer 3 Report

The manuscript is greatly improved.  There still some minor writing/editing issues that should be addressed.  These suggestions/comments are below.  

Line 87, "Healthcare workers are considered as the ...". 

Line 120, "...childhood vaccinations of HCWs at the ...". 

Line 133, "...workers, have USM staff emails. Malaysian ...". 

Line 140, "80% and the minimum sample size is ...". 

Line 204, "most respondents agreed that the polio vaccine ...". 

Line 207, "...with a good knowledge, as female respondents ...". 

Line 210, "(degree and above), the participants had significantly ...". 

Line 234, "...vaccination.  Healthcare workers with a good level...". 

Line 238, "...A substantial proportion of HCWs displayed ...". 

Line 240, "...from the participants. A majority agreed ...". 

Line 242, "...given to children.  Healthcare workers with a higher level ...". 

Line 246, "It is expected that the score of ...". 

Lines 249-250, "...of the respondents were not directly involved in ...". 

Line 252, "...practitioners and do not have enough exposure ...". 

Lines 263 & 264, "...after analyzing the data with multiple logistic regression, we already adjusted ...". 

Line 270, "...vaccinations.  Healthcare workers play a crucial role in ...". 

Line 287, "...effort should be taken into an action to strengthen ...". 

Line 292, "...to be considered as severe enough to make vaccines ...". 

Line 303, "vaccine safety, efficacy and to increase confidence ...". 

Lines 312-313, "...most countries, including Malaysia, have switched the ...".Line 318, "...confidence in vaccinations and preventing ...". 

Line 325, "...proportion of HCWs displayed a good ...". 

Line335, "...compared with non-healthcare professionals [36].  This ...".  Line 337, "..of vaccinations for a better attitude, in our case, on childhood vaccination.  Again, ...". 

Line 346, "...are the probable causes of the low ...". 

Lines 353 & 254, "...of HCWs from personal and professional perspectives and may be ...". 

Line 358, "...dangers associated with a vaccination are ...". 

Line 362, "...data on the effects of vaccinations, particularly accurate ...

"Line 364, "...of this study validated the Malay version ...". 

Line 374, "...limitation is that the questionnaire was ...". 

Line 375, "...based on the participant's self-administered, not to ...". 

Line 379, "...location mainly consisting of the Malay ...". 

Line 385, "...pandemic with the majority of the staff ...". 

Line 386, "...might not be representative of the ...". 

Line 388, "...and female respondents, as some limitations ..." 

Line 394, "...factor for poor knowledge which resulted in a negative ...". 

Lines 397 & 398, "...knowledge and to a have a positive attitude toward children vaccination among  ...". 

Author Response

Dear sir,

Thank you very much for your comments. We have made the corrections as suggested

No.

Comments

Correction

1

Line 120, "...childhood vaccinations of HCWs at the ...".

Line 133, "...workers, have USM staff emails. Malaysian ...".

Line 140, "80% and the minimum sample size is ...".

Correction made as per comment

2

Line 204, "most respondents agreed that the polio vaccine ...".

Line 207, "...with a good knowledge, as female respondents ...".

Line 210, "(degree and above), the participants had significantly ...".

Correction made as per comment

3

Line 234, "...vaccination.  Healthcare workers with a good level...".

Line 238, "...A substantial proportion of HCWs displayed ...".

Line 240, "...from the participants. A majority agreed ...".

Correction made as per comment

4

Line 242, "...given to children.  Healthcare workers with a higher level ...".

Line 246, "It is expected that the score of ...".

Lines 249-250, "...of the respondents were not directly involved in ...".

Correction made as per comment

5

Line 252, "...practitioners and do not have enough exposure ...".

Lines 263 & 264, "...after analyzing the data with multiple logistic regression, we already adjusted ...".

Line 270, "...vaccinations.  Healthcare workers play a crucial role in ...".

Correction made as per comment

6

Line 287, "...effort should be taken into an action to strengthen ...".

Line 292, "...to be considered as severe enough to make vaccines ...".

Line 303, "vaccine safety, efficacy and to increase confidence ...".

Correction made as per comment

7

Lines 312-313, "...most countries, including Malaysia, have switched the ...".Line 318, "...confidence in vaccinations and preventing ...".

Line 325, "...proportion of HCWs displayed a good ...".

Line335, "...compared with non-healthcare professionals [36].  This ...".  Line 337, "..of vaccinations for a better attitude, in our case, on childhood vaccination.  Again, ...".

Correction made as per comment

8

Line 346, "...are the probable causes of the low ...".

Lines 353 & 254, "...of HCWs from personal and professional perspectives and may be ...".

Line 358, "...dangers associated with a vaccination are ...".

Correction made as per comment

9

Line 362, "...data on the effects of vaccinations, particularly accurate ...

"Line 364, "...of this study validated the Malay version ...".

Line 374, "...limitation is that the questionnaire was ...".

Correction made as per comment

10

Line 375, "...based on the participant's self-administered, not to ...".

Line 379, "...location mainly consisting of the Malay ...".

Line 385, "...pandemic with the majority of the staff ...".

Correction made as per comment

11

Line 386, "...might not be representative of the ...".

Line 388, "...and female respondents, as some limitations ..."

Line 394, "...factor for poor knowledge which resulted in a negative ...".

Lines 397 & 398, "...knowledge and to a have a positive attitude toward children vaccination among  ...".

Correction made as per comment
